# Investigation and Biological Assessment of *Rumex vesicarius* L. Extract: Characterization of the Chemical Components and Antioxidant, Antimicrobial, Cytotoxic, and Anti-Dengue Vector Activity

**DOI:** 10.3390/molecules27103177

**Published:** 2022-05-16

**Authors:** Salama A. Salama, Zarraq E. AL-Faifi, Mostafa F. Masood, Yasser A. El-Amier

**Affiliations:** 1Biology Department, Faculty of Science, Jazan University, P.O. Box 114, Jazan 45142, Saudi Arabia; sasalama@jazanu.edu.sa (S.A.S.); mmasood@jazanu.edu.sa (M.F.M.); 2Center for Environmental Research and Studies, Jazan University, P.O. Box 2097, Jazan 42145, Saudi Arabia; zalfifi@jazanu.edu.sa; 3Department of Botany, Faculty of Science, Mansoura University, Mansoura 35516, Egypt

**Keywords:** *R. vesicarius*, GC–MS, chemical components, biological activity, HepG2, *Aedes aegypti*

## Abstract

The objective of this study was to assess the biological potency and chemical composition of *Rumex vesicarius* aboveground parts using GC–MS. In this approach, 44 components were investigated, comprising 99.99% of the total volatile compounds. The major components were classified as fatty acids and lipids (51.36%), oxygenated hydrocarbons (33.59%), amines (7.35%), carbohydrates (6.06%), steroids (1.21%), and alkaloids (0.42%). The major components were interpreted as 1,3-dihydroxypropan-2-yl oleate (oxygenated hydrocarbons, 18.96%), ethyl 2-hydroxycyclohexane-1-carboxylate (ester of fatty acid, 17.56%), and 2-propyltetrahydro-2*H*-pyran-3-ol (oxygenated hydrocarbons, 11.18%). The DPPH antioxidant activity of the extracted components of *R. vesicarius* verified that the shoot extract was the most potent with IC_50_ = 28.89 mg/L, with the percentages of radical scavenging activity at 74.28% ± 3.51%. The extracted plant, on the other hand, showed substantial antibacterial activity against the diverse bacterial species, namely, *Salmonella typhi* (23.46 ± 1.69), *Bacillus cereus* (22.91 ± 0.96), *E. coli* (21.07 ± 0.80), and *Staphylococcus aureus* (17.83 ± 0.67). In addition, the extracted plant was in vitro assessed as a considerable anticancer agent on HepG2 cells, in which MTT, cell proliferation cycle, and DNA fragmentation assessments were applied on culture and treated cells. The larvicidal efficacy of the extracted plant was also evaluated against *Aedes aegypti*, the dengue disease vector. As a result, we may infer that *R. vesicarius* extract increased cytocompatibility and cell migratory capabilities, and that it may be effective in mosquito control without causing harm.

## 1. Introduction

Wild plants are considered pharmacologically as a source of bioactive molecules. Early people treated their ailments using plants; hence, the history of medicinal plants is as long as the history of mankind [1,2]. About 80% of the world’s population, primarily in Africa and other underdeveloped countries, still depends mainly on traditional medicine for disease treatment. African medicinal plants have a vast range of biological qualities that need to be identified, recorded, and researched. Over 1300 medicinal plants are used in European countries, out of which 90% are wild [3]. These confidential pharmaceutical drugs accessible from natural sources have fewer unfavorable side-effects than synthetic sources [4].

Egypt’s flora includes around 2080–2094 species of seed plants and vascular cryptogams [5,6]. The family Polygonaceae is a temperate-zone cosmopolite. Around the globe, there are 46 genera and 1100 species. This family is represented in Egypt by 28 species under eight genera [5,7]. The plants of genus *Rumex* comprise almost 150 annual, biennial, and perennial species broadly distributed in temperate climates worldwide [7,8]. *Rumex* is a common household herb, which is beneficial in the treatment of constipation. It has cleaning effects and can be used to treat skin disorders [9,10]. *Rumex* spp. are reported to have hepatoprotective, phytotoxic, antioxidant, and antimicrobial properties [11,12,13,14,15].

The properties of natural substances from plants such as phenolics, flavonoids, and terpenoids have been broadly described. *Rumex vesicarius* L. (Hammeidda) is an annual semi-succulent pale green leafy herb with 15–30 cm height [7,16]. This herb, which is endemic to northern Africa and Asia [17], grows yearly throughout the wet seasons of autumn and spring. Many studies have shown that it is regarded to be a nutritious supplement plant due to its high levels of β-carotenes [18], vitamins (especially vitamin C), lipids, proteins, and organic acids. It also includes minerals such as potassium, sodium, calcium, magnesium, iron, manganese, and copper [16,19,20]. Recent studies have concentrated on the characterization of active chemical components, as well as biological uses of extracted plants. The whole plant is medicinally significant and heals a variety of ailments such as tumors, hepatic illnesses, poor digestion, constipation, calculi, heart problems, aches, spleen diseases, and hiccough [21,22].

*Aedes aegypti* is a mosquito that can spread dengue fever, chikungunya, Zika fever, Mayaro, and yellow fever viruses, as well as other disease agents. The continuous application of synthetic insecticides causes the development of resistance in vector species, biological magnification of toxic substances through the food chain, and adverse effects on environmental quality and nontarget organisms including human health [23,24].

Previous phytochemical investigation revealed the presence of phenolics, flavonoids anthraquinones, sesquiterpenes, and minerals [15,25,26], However, the anti-dengue vector activity of *R. vesicarius* extract was not previously studied. Despite its significance, there have been little studies on the plant. Therefore, the current study sought to analyze the chemical ingredients of the Egyptian ecospecies of *Rumex vesicarius* methanol extract using GC–MS spectroscopy analysis in order to investigate the biochemical components responsible for the biological effects. The work was continually extended in order to investigate the antioxidant properties of the extracted plant using the DPPH free-radical scavenging test, while the antibacterial and cytotoxic properties of the extracted plant were examined in vitro against several bacterial species and cancer cell lines. Furthermore, the larvicidal activity against *Aedes aegypti* was assessed.

## 2. Results and Discussion

### 2.1. Chemical Characterization of R. vesicarius Extract

The chemical components and structures of the methanol extract of *R. vesicarius* shoot were estimated by gas chromatography–mass spectrometry analysis (GC–MS) (Figure 1). The results in Table 1 state that the methanol extract contained mainly 44 components, which were categorized under numerous classes of compounds. Thus, the obtained results revealed that the chemical components of the extract of *R. vesicarius* constituted 99.99% of the chemical composition of the analyzed sample. The *R. vesicarius* extract had numerous main components (>2%) such as 2-propyltetrahydro-2*H*-pyran-3-ol, ethyl 2-hydroxycyclohexane-1-carboxylate, 1,3-dihydroxypropan-2-yl oleate, *N*2,*N*4-diisopropyl-6-(methylsulfonyl)-1,3,5-triazine-2,4-diamine, (*E*)-octadec-13-enoic acid, 2-hydroxypropane-1,3-diyl dipalmitate, high oleic safflower oil, 2,3-dihydroxypropyl palmitate, methyl octadeca-8,11-diynoate, (*E*)-(2-(chloroimino)-3-methylbutanoyl)-l-valine, methyl 5-((1*R*,2*R*)-2-undecylcyclopropyl)-pentanoate, and (2-phenyl-1,3-dioxolan-4-yl)methyl oleate. These compounds represented 80.39% of the total identified compounds (Table 1). The chemical composition of the methanolic extract of *R. vesicarius* in the present study is different from that reported in other studies on the Iraq ecospecies [27].

As indicated in Figure 2, the chemical constituents of the *R. vesicarius* methanol extract were divided into six categories: oxygenated hydrocarbons (33.59%), fatty acids and their derivatives (51.36%), amines (7.35%), carbohydrates (6.06%), steroids (1.21%), and alkaloids (0.42%). Therefore, the class of fatty acids and their derivatives, i.e., “lipids”, was the major class constituting 51.36%, and only 19,20-didehydroyohimbinone was identified as an indole alkaloid compound with 0.42% of the total methanol extract. High oleic safflower oil basically consisted of palmitic acid (4–8%), stearic acid (1–3%), oleic acid (70–80%), and linoleic acid (12–16%).

The documented amino acids included (*E*)-(2-(chloroimino)-3-methylbutanoyl)-l-valine (3.84%), and glutamic acid (0.34%), along with steroids such as estra-1,3,5(10)-trien-17β-ol (0.34%) and ethyl iso-allocholate (0.87%). 19,20-Didehydroyohimbinone was classified as an indole alkaloid and was interpreted with a composition percentage of 0.42% with a retention time of 7.88 min. Carbohydrates were the second most common class of the identified chemical components such as d-Gala-l-ido-octonic amide (0.26%), desulfosinigrin (0.27%), l-Gala-l-ido-octose (1.09%), melezitose (0.34%), d-ribo-hexos-3-ulose (0.30%), and methyl *N*-acetyl-d-glucosamide (0.14%).

### 2.2. Biological Characteristics of the Plant Extracts

#### 2.2.1. Antioxidant Activity—DPPH Assay

The antioxidant activity was evaluated for the methanol extract of *R. vesicarius* using the DPPH colorimetric assay. Ascorbic acid was used as the reference standard for a comparison of the obtained results. The investigated extract showed potent antioxidant activities as compared to the results of ascorbic acid (IC_50_ = 12.48 mg/L). Subsequently, the results, as presented in Table 2, verified that the shoot extract had the most potent antioxidant scavenging activity with IC_50_ = 28.89 mg/L. The prevalence of fatty acids and their derivatives (51.36%), as well as oxygenated hydrocarbons (33.59%), was the major factor controlling the mechanism of the reactions involved in the evaluation of the antioxidant aptitude of the investigated extracts. The present results of *Rumex vesicarius* are in agreement with the results of Nishina et al. [28], Demirezer et al. [29], Al-Ismail et al. [30], Özen [31], and Li and Liu [32].

Fatty acids and lipids isolated from *Chenopodium ambrosioides* and *Euphorbia lathyrus* displayed potent antioxidant activities for scavenging free radicals in solution [33,34]. The major compounds 2-propyltetrahydro-2*H*-pyran-3-ol, ethyl 2-hydroxycyclohexane-1-carboxylate, and 1,3-dihydroxypropan-2-yl oleate (11.18%, 17.56%, and 18.96%, respectively) act as substantial antioxidant agents. Many factors influence the mechanism of antioxidant activity; in general, the antioxidant capacity of bioactive compounds is determined by the ability of reactive oxygen species, such as phenolics, fatty acids, terpenes, oxygenated hydrocarbons, or carbohydrates, to scavenge or stabilize free radicals [35,36,37]. In this study, the shoot extract outperformed the other wild plant extracts in terms of antioxidant activity. Furthermore, antioxidants mainly come from plants in the form of phenolic compounds such as flavonoids, phenolic acids, ascorbic acid, and carotenoids. According to our previous study, this plant contains nonvolatile compounds such as flavonoids and phenolics [15,22].

The chemical constituents of plant extracts also tend to form a complex with DPPH solution and accordingly stabilize or scavenge the free radicals in solution [38,39,40]. The antioxidant activity was attributed to the number of free hydroxyl groups [41,42,43].

#### 2.2.2. Antibacterial Activity

The antibacterial potential of the extracted *R. vesicarius* was assessed using the agar disc diffusion assay [43]. In this course, four Gram-negative and four Gram-positive bacterial strains were selected to estimate the antimicrobial effects of the investigated extracts. Accordingly, the extracted sample and standard antibiotics were prepared in a concentration of 10 mg/L (Table 3). The results indicated that the extract revealed potent antibacterial activities toward diverse bacterial species. The *S.* Typhimurium was the mainly influenced bacterial strain, while *P. aeruginosa* and *S. xylosus* were the main resistant species (Table 3). The tested organisms could be ordered according to their sensitivity as follows: *S.* Typhimurium < *B. cereus* < *E. coli* < *S. aureus* < *K. pneumoniae* < *S. haemolyticus* < *S. xylosus* < *P. aeruginosa*. The standard antibiotics, ampicillin, azithromycin, cephradin, and tetracycline, showed variable activities. *P. aeruginosa* was entirely resistant to ampicillin, cephradin, and tetracycline, while *S.* Typhimurium was resistant to ampicillin, azithromycin, cephradin, and penicillin (Table 3). Moreover, the plant extract revealed potent antibacterial activity toward *Escherichia coli*, *Salmonella* Typhimurium, and *Bacillus cereus* species (21 ± 0.80, 23.46 ± 1.69, and 22.91 ± 0.96 mm), but had less potency toward *P. aeruginosa* and *S. xylosus* compared to the antibiotics.

The methanol extract of *R. vesicarius* in the present study showed about threefold greater activity toward *S. aureus, K. pneumoniae,* and *E. coil* than the Indian variety [44,45]. Several experiments on different plant parts of different species of *Rumex* confirmed their potent antibacterial activity toward both Gram-positive and Gram-negative bacteria [15,46,47,48]. These differences in the antimicrobial potential could be attributed to environmental conditions such as climate and soil type, which can influence the extract components [49].

The methanol extract of *R. vesicarius* appeared to exhibit potent antibacterial activity regardless of whether species were Gram-negative or Gram-positive species. Hence, the assorted results achieved for the same extract were a function of its numerous components. Thus, the composition of the extract was evaluated to determine the effectiveness of specific compounds against bacterial species. In particular, by studying the role of the main chemical components of the extracted *R. vesicarius* as antibacterial agents, we found that terpenes, oxygenated hydrocarbons, and carbohydrates presented higher antibacterial potential [50,51,52]. Moreover, fatty acids and lipids in the essential oils of some extracted plants display improved antimicrobial characteristics [53,54,55]. This plant contains nonvolatile chemicals such as flavonoids and phenolics, according to our study [15]. These chemicals showed antibacterial efficacy against bacteria with multiple resistances [45,47].

#### 2.2.3. Cytotoxicity and Cell Migration Analysis

Recently, studies have been conducted for the development of novel treatment protocols using herbs in the creation of novel treatment approaches for numerous cancer varieties [56]. Moreover, the utility of herbal extracts with characteristic therapeutic properties is beneficial in pharmaceutical chemistry to achieve increased efficiency and biological potency [57]. Therefore, the utility of plant extracts is predicted to result in reduced side-effects compared to chemotherapeutic drugs [58]. Herein, *R. vesicarius* extract was assessed for antitumor activity using the MTT assay. This assay fits cell numbers to a growth curve. The MTT experiments were performed in the dark. Hepatocellular carcinoma (HepG2) cells were chosen to assess the antitumor activity of *R. vesicarius* extract. The IC_50_ (half-maximal inhibitory concentration) values were obtained through plotting the cell viability against drug concentration (µM). Thus, the cytotoxic potential is inversely proportional to the used concentrations and IC_50_ values. A control sample of MTT solution was seeded in one well. Serial dilutions of five extract concentrations were used (31.3, 62.5, 125, 500, and 1000 µg/mL).

The half-maximal inhibitory concentrations (IC_50_, µg/mL) and the absorbance reads for each sample concentration are shown in Table 4. The results revealed that the shoot extract possessed weaker cytotoxic activity with an IC_50_ value at 352.4 µg/mL. The mode of action of cytotoxicity is dependent on the phytochemical composition, concentration, and nature of cancer cell lines [59], in addition to the surface morphology, size, and aggregation of the extract particles. Furthermore, compounds such as polyphenols and flavonoids may be responsible for the anti-inflammatory and cytotoxic properties of plant extracts [15,60]. According to our investigation [15], *R. vesicarius* is composed of nonvolatile compounds including flavonoids and phenolics.

Recently, cytotoxicity, apoptosis, proliferation, cell-cycle progression, genetic damage, and apoptotic cell death were reported in HepG2 cancer cells [60,61,62]. Herein, the improved cell migration properties of *R. vesicarius* extract were also estimated using a cell motility assay [63,64,65]. The films revealed perfect wound healing activity in HepG2 cell lines. The cell lines of HepG2 migrated to the center of the scratch, covering the main part of the scratch after staining. The cell migration exemplified by microscopic images at definite intervals of time is presented in Figure 3. The plant extracted displayed no migration at 0 h or after 26 h of incubation. In contrast, after staining, the images showed a better healing rate at the varying doses of the extracted plant with a much faster healing rate upon increasing the concentration of the investigated extract. Consequently, the maximum healing rate was noted when using the plant extract at the IC_50_ dose after staining. The migration of cells from the edges to the center of the scratch was achieved after diffusion of the extract into the generated scratch, thus closing the scratch with a higher rate of migration. The cell migration activity for wound healing is attributed to the effect of several components of the plant extract, e.g., minerals, vitamins, activated enzymes, polyphenolics, and polysaccharides [66]. The impact of these components improves the healing of wounds through increased growth factors, angiogenesis, and reduced aggregation of platelets, while serving as exceptional scavengers for free radicals t the wound site [67]. The induced cell growth results from the impact of phytochemicals, e.g., reactive oxygen species, accompanied by respectable antioxidant potency and improved wound closure activity. Accordingly, on the basis of the substantial therapeutic value of the extracted *R. vesicarius*, it can be considered as a proficient biocompatible material for wound dressing; future in vivo studies and clinical analysis are required to confirm its practical applicability.

##### DNA Fragmentation

DNA fragmentation was used as an indicator of apoptosis for the investigation of apoptotic cells. Oxygenated hydrocarbons display strong antioxidant effects and, accordingly, can protect the cellular DNA from oxidative damage in many degenerative diseases associated with oxidative stress [68]. Thus, the occurrence of DNA fragmentation, to estimate the proapoptotic effect of *R. vesicarius*, was discovered in all of the intended groups using the DNA gel electrophoresis procedure. The densitometry analysis of the treatment-dependent DNA laddering effect identified a noteworthy increase in the percentage of DNA fragmentation (MDR1: 40.23%; CD44: 70.53%) as compared to the untreated control groups (Figure 4). According to the results, the methanol extract of *R. vesicarius* could change the cell morphology, thereby inhibiting cell growth in a time- and dose-dependent manner, leading to DNA degradation.

##### The EC_50_ Value of *R. vesicarius* Extract

The dose–response relationship is plotted in Figure 5. The dose–response curve in Figure 5a was normalized along the *X*-axis by its EC_50_ value (Figure 5b). The IC_50_ of the *R. vesicarius* extract was estimated by plotting the sample absorbance against the log of doses at different concentrations of the serial dilution (Figure 5). The response increased upon increasing the extract concentration, with the vertical point of the curve revealing the EC_50_ value [69,70]. The data analysis showed that a dose of 500 µg/mL had a cytotoxic influence on HepG2 cell lines.

#### 2.2.4. Larvicidal Bioassay

Data given in Table 5 indicate the larvicidal activity of the methanolic extract of *R. vesicarius* against the third-instar larvae of *Ae. aegypti* at 24 and 48 h post treatment. The results revealed a significant increase in larval mortality with all concentrations of the methanolic extract of *R. vesicarius*. The maximum larval mortality was recorded at concentrations of 1200 mg/L after 24 and 48 h (28.9% and 42.6%, respectively), while the lowest mortality was 7.7% after 24 h when the concentration was 250 mg/L and 1.1% after 48 h when the concentration was 125 mg/L. The LC_50_ (the concentration of extract that causes 50% mortality) of the extract was 19.99 and 14.97 mg/L, while the LC_90_ (the concentration of extract that causes 90% mortality) was 36.12 and 27.43 mg/L after 24 and 48 h, respectively. The toxicity of plants depends on their various bioactive compounds, including phenolics, terpenoids, flavonoids, and alkaloids, either as single or joint compounds [71]. Moreover, desert plants such as *R. vesicarius* are easily available and a rich source of bioactive compounds, which function as larvicides, pupicides, or mosquito repellents [15,72].

These results are consistent with those obtained by Nasir et al. [71]*,* who used essential oils from some medicinal plants against *Ae. albopictus*, where ginger was revealed to be most effective with the lowest LC_50_ values after 8 and 16 h, followed by peppermint, basil, eucalyptus, and neem. Ullah et al. [72] recorded LC_50_ and LC_90_ values of 50.27 and 203.99, and 17.77 and 206.49 mg/L for *Cassia fistula* and *Nicotiana tabacum* extracts against larvae of *C. quinquefasciatus*. Hassanain et al. [73] used a petroleum ether extract from leaves of *Lantana camara* against larvae of *An. multicolor*, recording the highest larval mortality (100.0%) at 140 mg/L. Shehata [74] found that a petroleum ether extract from leaves of *Prunus domestica* and *Rhamnus cathartica* was more effective against *C. pipiens* (LC_50_: 33.3 and 63.4 mg/L, respectively) than chloroform (LC_50_: 70.8 and 192.1 mg/L) and methanolic (LC_50_: 132.7 and 273.5 mg/L, respectively) extracts. Dey et al. [75] found that the aqueous extract of *Piper longum* showed the highest larval mortality after 24 h. of treatment against *Ae. aegypti*, *An. Stephensi*, and *C. quinquefasciatus*. Farag et al. [76] observed that the peel powder of *Punica granatum*, extracted with petroleum ether, had potential toxicological effects against third-instar larvae of *C. pipiens* with an LC_50_ value of 95.66 mg/L.

## 3. Materials and Methods

### 3.1. Plant Material and Extraction Process

The aerial parts of *Rumex vesicarius* were acquired from Wadi Hagoul, north of the Eastern Desert, Egypt (29°54′47.46″ N 32°12′28.10″ E) during the flowering season (April 2021) from different populations. The plant was identified according to Tackholm [5] and Boulos [7]. A voucher specimen (Mans. 0161822009) was prepared and deposited in the Herbarium of Botany Department, Faculty of Science, Mansoura University, Egypt.

The collected samples were cleaned and naturally dried. Then, 10 g of dried plant materials were placed in a 250 mL conical flask containing 150 mL of methanol. The mixture was then transferred to a water bath shaker (Memmert WB14, Schwabach, Germany), with continuous shaking for 2 h at room temperature. Whatman filter papers (No. 1, 125 mm, Cat No. 1001 125, Germany) were used for the filtration step of the mixture. The final concentrations of the extracted plant issues were investigated, and the extracts were stored at 4 °C [77].

### 3.2. Gas Chromatography–Mass Spectrometry Analysis (GC-MS)

The chemical composition of the extracted *R. vesicarius* plant was characterized by implementing the plant extract on a Trace GC-TSQ mass spectrometer (Thermo Scientific, Austin, TX, USA) with a direct capillary column TG-5MS (30 m × 0.25 mm × 0.25 μm film thickness) [78]. The temperature of the column oven was firstly held at 50 °C, raised subsequently by a rate of 5 °C per minute to reach 250 °C, and held for 2 min, before raising the temperature to the final temperature (300 °C) by 30 °C per minute and holding for 2 min. The injector and MS transfer line temperature were kept at 270 and 260 °C, respectively. Helium (He) was used as the inert carrier gas at a constant flow rate of 1 mL/min. The solvent was released after 4 min, and diluted samples of 1 μL were injected directly using an Autosampler AS1300 coupled with GC in split mode. EI mass spectroscopy was collected at 70 eV ionization voltage over a range of 50–500 for *m/z* in packed scan mode. The temperature of the ion source was fixed at 200 °C. The chemical components of the individual extracted plant materials were interpreted through a comparison of their mass spectral data with those in the WILEY 09 and NIST 14 mass spectrometry databases. The GC–MS analysis indicated five conceivable components for each significant peak. In the case of dissimilar proposed components, the factors of probability and the fragmentation patterns of the main structure were used to determine the selected structure.

### 3.3. Antioxidant DPPH Assay

A stock solution of extracted *R. vesicarius* was diluted in methanol to the desirable concentrations (5, 10, 20, 30, 40, and 50 mg/L). DPPH solution (1 mL, 0.135 mM) was added to each prepared concentration of sample solution. Catechol was used as a standard with the same concentrations as the tested samples. The samples were kept at room temperature in the dark for 30 min, and the absorbance was measured at λ = 517 nm using a UV/Vis spectrophotometer (Spekol 11 spectrophotometer, analytic Jena AG, Jena, Germany). The percentages of antioxidant scavenging activities were calculated using the following equation, in which the DPPH solution in methanol was used as a control:(1)$$\%\;\mathrm{Inhibition}=\frac{A\;\mathrm{control}\;-\;A\;\mathrm{sample}}{A\;\mathrm{control}}\times 100.$$

The procedure was applied following previous reports [79,80] with insignificant modifications. The inhibitory concentrations (IC_50_, mg·L^−1^) were calculated by applying the plotted exponential curve [81], which specified the relationship of the sample concentration with the percentage of remaining DPPH^•^ radicals.

### 3.4. Assessment of the Antibacterial Activity

*Culture media*: The nutrient agar (28.0 gm) was placed in a conical flask (2 L) and mixed with 1 L of distilled water. The contents of the conical flask were heated until boiling point for the complete dissolution of the medium. Sterilization of the mixture by autoclave was accomplished at 15 lb pressure, 121 °C for 15 min. The medium was left to cool to 45–50 °C, and then enriched with 5–10% blood or biological fluids. The mixture was shaken and poured into sterile Petri plates.

*Bacterial species*: The microbial species were purchased from the Cairo Microbiological Resources Center (Cairo MIRCEN), Faculty of Agriculture, Ain Shams University. The Gram-negative bacteria used were as follows: *Escherichia coli* (ATCC 10536), *Pseudomonas aeruginosa* (ATCC 9027), *Salmonella* Typhimurium (ATCC 25566), *Klebsiella pneumoniae* (ATCC 10028). The Gram-positive bacteria used were as follows: *Bacillus cereus* (EMCC number 1080), *Staphylococcus aureus* (ATCC 6538), *Staphylococcus haemolyticus* (ATCC 29970), and *Staphylococcus xylosus* (NCCP 10937). Cephradin, tetracycline, azithromycin, and ampicillin were used as standard antibiotics.

*Microbial testing:* The antimicrobial activity of the extracted plant issues was estimated using an agar well diffusion assay [82] with inocula containing 10^6^ bacterial cells/mL to spread on nutrient agar plates. The sterilized filter paper discs (Whatman No.1, 6 mm in diameter) were immersed overnight in the extracted issues of the plant until saturation, and another set of filter paper discs were immersed in methanol as a control. The discs were placed on the surface of agar plates seeded with definite bacterial microorganisms. The plates were incubated at 37 °C for 18–24 h, and the inhibition zone diameters (mm) were measured [83].

### 3.5. Cytotoxicity and Cell Proliferation

*Tumor cell lines:* HePG-2 (hepatocellular carcinoma) cell lines were purchased from ATCC (VACSERA, Cairo, Egypt).

*Preparation of MTT solution:* The MTT (3-(4,5-dimethyl-2-thiazolyl)-2,5-diphenyl-2*H*-tetrazolium bromide) solution was prepared by mixing a solution of MTT in water (10 mg/mL), ethanol (20 mg/mL), and buffered salt solutions and media (5 mg/mL). The mixture was mixed by vortex or sonication, filtered, and then stored at −20 °C.

The MTT assay was used for evaluation of the extracted *R. vesicarius* [84,85]. HepG2 cell lines were seeded at a concentration of 3 × 10^3^ cells/well suspended in 100 μL of complete medium in 96-well plates, and the experiments were implemented in duplicates. Subsequently, the plates were incubated for 24 h in 5% CO_2_ at 37 °C until settling and adherence. The plant extract was prepared in serial dilutions (31.3, 62.5, 125, 500, and 1000 µg/mL) and applied to cell lines for 48 h. The medium was removed by aspiration, and MTT (0.5 mg/mL), dissolved in the culture medium, was added onto cells and incubated at 37 °C and 5% CO_2_ for 4 h. SDS (100 µL) was added to each well, and the cell growth was measured at λ_max._ = 570 nm (BioTek, Elx800, Winooski, VT, USA), expressing the results as a percentage of the control. The IC_50_ values were estimated using Origin 8.0^®^ software (Origin Lab Corporation, Northampton, MA, USA). The IC_50_ values were estimated using the equation IC_50_ = (0.57−b)/a. The inhibition percentage was estimated using the following equation (Equation (2)):(2)$$\%\;\mathrm{Inhibition}=\frac{{\mathrm{Absorbance}}_{\mathrm{control}\;}–{\mathrm{Absorbance}}_{\mathrm{control}\;}}{{\mathrm{Optical}\;\mathrm{density}}_{\mathrm{control}\;}}\times 100.$$

The relative cell viability percentage was estimated using the following equation:(3)$$\%\;\mathrm{Cell}\;\mathrm{viability}=\frac{{\mathrm{Absorbance}}_{\mathrm{samples}\;}–{\mathrm{Absorbance}}_{\mathrm{blank}\;}}{{\mathrm{Optical}\;\mathrm{density}}_{\mathrm{control}\;}–{\mathrm{Absorbance}}_{\mathrm{blank}\;}}\times 100.$$

### 3.6. Cell Motility Assay

Cells were seeded in a six-well plate. The monolayer cells were scratched using a 10 µL pipette tip to create a wound, and cells were washed twice by PBS and then treated using the IC_5_, IC_10_, IC_25_, and IC_50_ of the *R. vesicarius* extract. The wounds were detected using phase-contrast microscopy on an inverted microscope. Images were taken regularly at 0 h, after staining, and after 26 h [86].

### 3.7. Conventional PCR

The conventional PCR technique was used for determination of the mRNA expression of CD44 and MDR1. The cells were treated with *R. vesicarius* extract for 24 h and collected to extract cellular RNA; then, total RNA was reverse-transcribed to cDNA (Qiagen, Germantown, MD, USA). The reaction included 1 µL of cDNA in a total volume of 20 μL containing 10 μL of Master Mix (Dream Taq Green PCR Master mix 2×, Thermo Fisher, Waltham, MA, USA), forward primer (0.5 μM), reverse primer (0.5 μM), and nuclease-free water. The thermal cycle conditions were as follows: 30 cycles of denaturation at 95 °C for 1 min, followed by annealing for 1 min at 72 °C, with a 10 min final extension at 72 °C. The sequences of the primers used for the determination of mRNA expression were as follows: MDR1 at 58 °C: 5′–CCC ATC ATT GCA ATA GCA GG–3′ (forward); 5′–TGT TCA AAC TTC TGC TCC TGA–3′ (reverse); CD44 at 55 °C: 5′–TTT GCA TTG CAG TCA ACA GTC–3′ (forward); 5′–TTA CAC CCC AAT CTT CAT GTC CAC–3′ (reverse).

### 3.8. Mosquitocidal Assay

#### 3.8.1. *Aedes aegypti* Colony

Larvae of *Ae. aegypti* were obtained from the Center for Disease Vector in Jizan, and they were reared for six generations at the Center of Environmental Research, Faculty of Science, Jazan University under controlled conditions (temperature: 27 ± 2 °C; RH: 70–80%; 12 h/12 h light–dark regime). Adult mosquitoes were kept in 30 × 30 × 30 cm wooden cages and were provided daily with cotton pieces soaked in 10.0% sucrose solution for a period of 3 days after emergence. After this period, females were allowed to take a blood meal from a pigeon host, which is necessary for laying eggs (anautogeny). A plastic cup (15 × 15 cm) containing dechlorinated tap water was placed in the cage for laying eggs. The resulting egg rafts were picked up from the plastic dish and transferred into plastic pans (25 × 30 × 15 cm) containing 3 L of tap water and left for 24 h. The hatching larvae were provided daily with a piece of bread as their diet. This diet was found to be the most preferable food for larval development and female fecundity [87].

#### 3.8.2. Larvicidal Activity of Tested Plant Extracts

For larvicidal activity, the tested material of methanol extracts was dissolved in 0.1 mL of methanol, while the tested material of acetone and chloroform extracts were dissolved in two drops of Tween-80 as emulsifier to facilitate the dissolving of tested material in water. Different ranges of concentrations of each extract were prepared in order to detect mortalities. All experiments with the tested materials were performed in 100 mL of dechlorinated tap water contained in 200 mL plastic cups. Then, third-instar larvae (10 larvae) were placed immediately into plastic cups containing different concentrations of extracts. Three replicates were usually used for each tested concentration. All plastic cups were incubated under controlled conditions of the mosquito colony, and mortality was subsequently recorded. Control larvae received 0.1 mL of methanol or two drops of Tween-80 in 100 mL of water. Mortality was recorded daily, and dead larvae and pupae were removed until adult emergence [87]. Larval mortality was indicated by a failure to respond to mechanical stimulation. The larval mortality percentage was estimated by using the following equation [88]:(4)$$\mathrm{Larval}\;\mathrm{mortality}\;=\frac{\mathrm{Number}\;\mathrm{of}\;\mathrm{dead}\;\mathrm{larvae}}{\mathrm{Number}\;\mathrm{of}\;\mathrm{treated}\;\mathrm{larvae}}\times 100.$$

### 3.9. Statistical Analysis

The antioxidant, antibacterial, and larvicidal activity studies were repeated three times with three replications, and the results were submitted to a one-way ANOVA to determine the significance of the differences between samples using the Costat software program (CoHort Software, Monterey, CA, USA).

## 4. Conclusions

In summary, 44 components were identified from the methanol extract of *R. vesicarius* shoots by GC–MS analysis. The majority of the characterized components were fatty acids and their derivatives, representing 51.36% of the total composition of the methanol extract. In addition, the major components were interpreted as 2-propyltetrahydro-2*H*-pyran-3-ol, ethyl 2-hydroxycyclohexane-1-carboxylate, and 1,3-dihydroxypropan-2-yl oleate (11.18%, 17.56%, and 18.96% of the composition). The *R. vesicarius* shoots displayed various biological properties such as antioxidant, antibacterial, and cytotoxic activities, as well as larvicidal activity against *Aedes aegypti*, the dengue disease vector, at low concentrations relative to the results of the respective standards. Precisely, the shoot extract exhibited the most potent antioxidant activity, indicating its higher potency to trap the free radicals in the DPPH solution. The extracted shoot of *R. vesicarius* revealed the most potent antibacterial activity against Gram-negative bacterial species, in which the extract showed broad-spectrum activity against *Escherichia coli*, *Salmonella* Typhimurium, and *Bacillus cereus*. The current study investigated the ability of the plant extract to improve the proliferation and viability of hepatocellular carcinoma cells in a wound closure in vitro assay. The remarkable percentages of terpenes, fatty acids, and their esters characterized from the *R. vesicarius* extract, as well as the considerable biological results, support the opportunity for future studies on this plant for drug discovery from a natural source. Furthermore, *R. vesicarius* extract appears to have great potential for controlling disease vector insects.

## Figures and Tables

**Figure 1 molecules-27-03177-f001:**
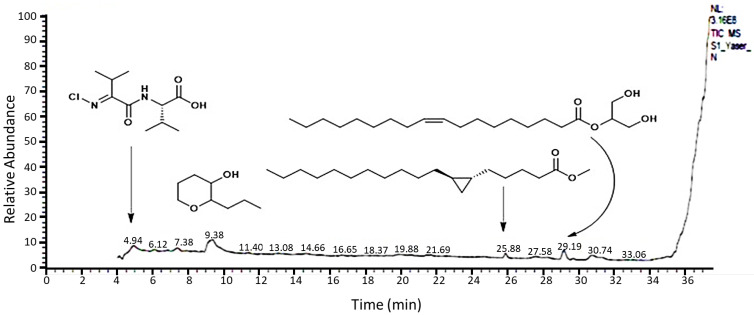
Chromatogram and structures of main components of the methanol extract of *Rumex vesicarius* shoot by GC–MS.

**Figure 2 molecules-27-03177-f002:**
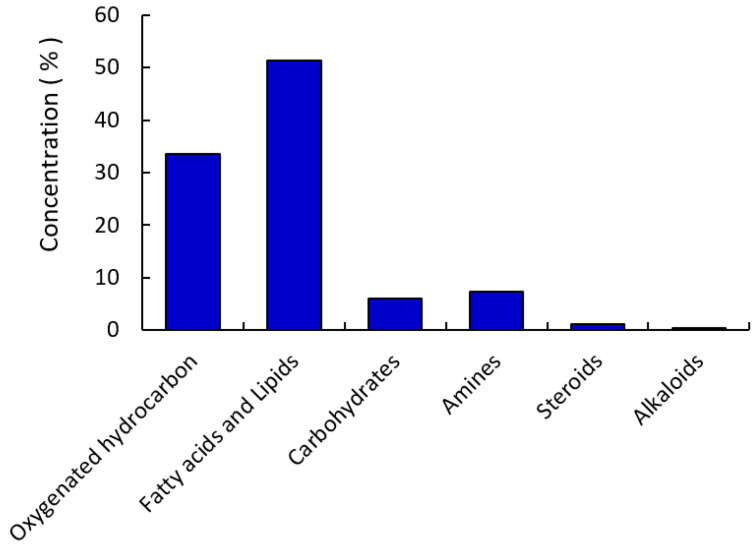
The categorized chemical constituents identified from the extracted *R. vesicarius* according to GC–MS analysis.

**Figure 3 molecules-27-03177-f003:**
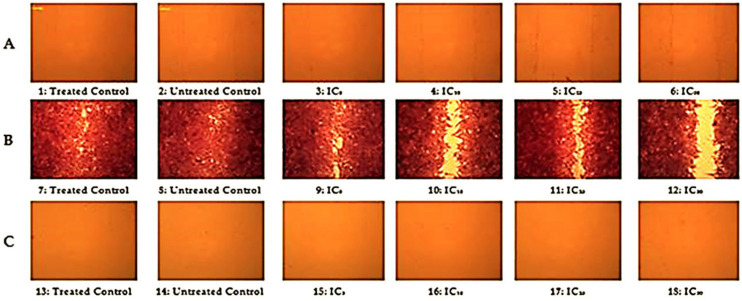
The impact of the methanol extract of *R. vesicarius* shoots on the cell-cycle progression of HepG2. (**A**) Microscopic reflection (20× magnification) of HepG2 cell line images at 0 h. (**B**) Microscopic reflection of cell line images after staining. (**C**) Microscopic reflection of cell line images after 26 h. The photomicrographs refer to the control groups and those treated at different doses of the plant extract at regular intervals.

**Figure 4 molecules-27-03177-f004:**
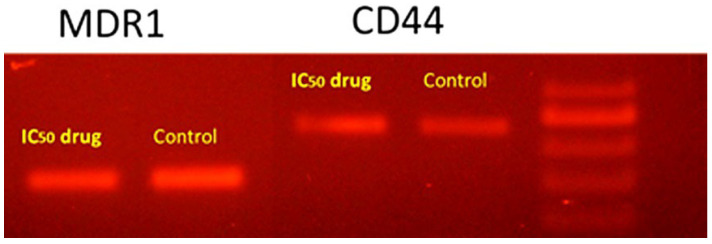
Fragmentation analysis of DNA specifying the noticeable effects of *Rumex vesicarius* in prompting a DNA laddering effect as a marker of cell apoptosis. Characterization of the MDR1 and CD44 antibodies as a target of *R. vesicarius-*dependent apoptosis.

**Figure 5 molecules-27-03177-f005:**
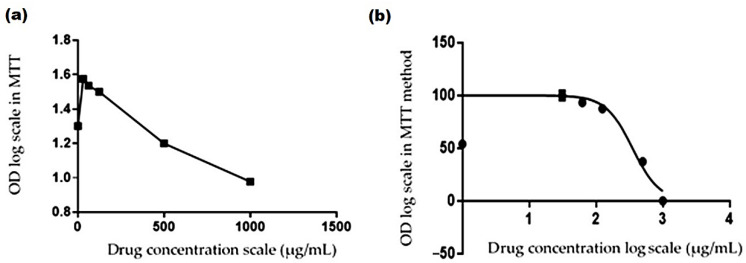
Dose-response curves of *R. vesicarius* extract in inhibiting the cell growth as a percentage of the control against HepG2 cell lines. (**a**) Transformation of the extracted *R. vesicarius*. (**b**) Normalization of the transformation of *R. vesicarius* extract. OD is the absorbance value at specific concentrations of the tested plant extract.

**Table 1 molecules-27-03177-t001:** The characterized chemical components isolated from the extracted shoots of *Rumex vesicarius*.

Entry	Chemical Name	Classification	Rt	MF	Composition %
Oxygenated Hydrocarbon					
1	(*Z*)-2-Ethylidene-1,5-dimethyl-3,3-diphenylpyrrolidine	Aryl substituted cyclic amine	4.21	C_20_H_23_N	0.92
2	3-(2-Oxocyclohexyl)propanenitrile	Oxygenated hydrocarbon	5.54	C_9_H_13_NO	0.41
3	Ethyl 2-hydroxycyclohexane-1-carboxylate	Oxygenated hydrocarbon	9.22	C_9_H1_6_O_3_	18.96
4	2-Propyltetrahydro-2*H*-pyran-3-ol	Oxygenated hydrocarbon	9.38	C_8_H_16_O_2_	11.18
5	Ascaridole epoxide	Oxygenated hydrocarbon	12.99	C_10_H_16_O_3_	0.67
6	3,5-Heptadienal, 2-ethylidene-6-methyl-“(2*Z*,3*E*)-2-ethylidene-6-methylhepta-3,5-dienal”	Oxygenated hydrocarbon	13.07	C_10_H_14_O	0.28
7	1,25-Dihydroxyvitamin D3, TMS derivative	Oxygenated hydrocarbon	34.16	C_30_H_52_O_3_Si	0.20
8	Deoxyspergualin	Polyamine spermidine	8.74	C_17_H_37_N_7_O_3_	0.36
9	Methyl 2,2,3,3,4,4,4-heptafluorobutanoate	Ester	12.87	C_5_H_3_F_7_O_2_	0.20
10	Tetraacetyl-d-xylonic nitrile	Polyester	14.63	C_14_H_17_NO_9_	0.41
Fatty Acids and Lipids					
11	High oleic safflower oil	Vegetable oil “80% oleic acid”	4.45	C_21_H_22_O_11_	2.93
12	Methyl octadeca-8,11-diynoate	Fatty acid	6.13	C_19_H_30_O_2_	3.82
13	(2-Phenyl-1,3-dioxolan-4-yl)methyl oleate	Fatty-acid derivative	7.40	C_28_H_44_O_4_	5.75
14	bis(2-Ethylhexyl) adipate	Ester of fatty acid	7.94	C_22_H_42_O_4_	0.80
15	d-Lyxo-d-manno-nononic-1,4-lactone	Lactone of tetronic acid	11.31	C_9_H_16_O_9_	0.56
16	*tert*-Butyl palmitate	Ester of fatty acid	14.52	C_20_H_40_O_2_	0.26
17	Oleic acid	Fatty acid	14.67	C_18_H_34_O_2_	0.57
18	9-Hexadecenoic acid	Fatty acid	15.20	C_16_H_30_O_2_	0.17
19	*trans*-2-Dodecenoic acid	Fatty acid	15.52	C_12_H_22_O_2_	0.24
20	2-Hydroxypropane-1,3-diyl (9*E*,9’*E*)-bis(octadec-9-enoate)	Diester derivative of fatty acid	16.64	C_39_H_72_O_5_	0.94
21	(*E*)-Octadec-13-enoic acid	Fatty acid	17.13	C_18_H_34_O_2_	2.74
22	2-Bromotetradecanoic acid	Fatty-acid derivative	19.89	C_14_H_27_BrO_2_	1.45
23	[1,1’-Bicyclopropyl]-2-octanoic acid, 2’-hexyl-, methyl ester	Fatty-acid derivative	19.96	C_21_H_38_O_2_	0.90
24	8-((2*R*,3*S*)-3-Octyloxiran-2-yl)octanoic acid	Fatty acid	21.41	C_18_H_34_O_3_	1.09
25	2,3-Dihydroxypropyl stearate	Fatty acid	21.69	C_21_H_42_O_4_	1.23
26	Methyl 5-((1*R*,2*R*)-2-undecylcyclopropyl)-pentanoate	Fatty-acid derivative	25.88	C_20_H_38_O_2_	4.50
27	2-Hydroxypropane-1,3-diyl dipalmitate	Ester of fatty acid	28.31	C_35_H_68_O_5_	2.87
28	1,3-Dihydroxypropan-2-yl oleate	Ester of fatty acid	29.18	C_21_H_40_O_4_	17.56
29	Methyl 11-((2*R*,3*R*)-3-pentyloxiran-2-yl)undecanoate	Ester of fatty acid	29.70	C_19_H_36_O_3_	1.75
30	9-Octadecenoic acid,1,2,3-propanetriyl ester, (*E*,*E*,*E*)-	Ester of fatty acid	31.35	C_57_H_104_O_6_	1.23
Carbohydrates					
31	d-Gala-l-ido-octonic amide “2,3,4,5,6,7,8-heptahydroxyoctanamide”	Carbohydrate amide	6.50	C_8_H_17_NO_8_	0.26
32	Desulfosinigrin “1-*S*-[(1*E*)-*N*-hydroxy-3-butenimidoyl]-1-thiohexopyranose”	Glycoside	6.64	C_10_H_17_NO_6_S	0.27
33	l-Gala-l-ido-octose	Carbohydrate	6.85	C_8_H_16_O_8_	1.09
34	Melezitose	Trisaccharide sugar	8.79	C_18_H_32_O_16_	0.34
35	d-Ribo-hexos-3-ulose “(2*S*,4*R*,5*R*)-2,4,5,6-tetrahydroxy-3-oxohexanal”	Dicarbonyl sugar	9.84	C_6_H_10_O_6_	0.30
36	2,3-Dihydroxypropyl palmitate	1-Monoacylglycerols	4.89	C_19_H_38_O_4_	3.80
Amines					
37	*N*2,*N*4-Diisopropyl-6-(methylsulfonyl)-1,3,5-triazine-2,4-diamine	Hetryl amine	4.14	C_10_H_19_N_5_O_2_S	2.44
38	(*E*)-(2-(Chloroimino)-3-methylbutanoyl)-l-valine	Amino acid	4.94	C_10_H_17_C_l_N_2_O_3_	3.84
39	*S*-(2-Aminoethyl) *O*-hydrogen sulfurothioate	Amino-thioester	11.39	C_2_H_7_NO_3_S_2_	0.59
40	Glutamic acid	Amino acid	11.46	C_5_H_9_NO_4_	0.34
41	Methyl *N*-acetyl-d-glucosamide	*N*-Acetyl-d-glucosamine	10.35	C_9_H_17_NO_6_	0.14
Steroids					
42	Estra-1,3,5(10)-trien-17β-ol	Steroid	20.54	C_18_H_24_O	0.34
43	Ethyl iso-allocholate	Steroidal ester	34.94	C_26_H_44_O_5_	0.87
Alkaloids					
44	19,20-Didehydroyohimbinone	Indole alkaloid	7.88	C_21_H_22_N_2_O_3_	0.42
	Total				99.99

RT: retention time, MF: molecular formula.

**Table 2 molecules-27-03177-t002:** Radical scavenging activity (%) and IC_50_ values (mg/L) at various concentrations of the methanol extract of *R. vesicarius* and the standard ascorbic acid according to DPPH assay.

Treatment	Conc. (mg/L)	Radical Scavenging Activity (%)	IC_50_ (mg/L)
*Rumex vesicarius* L	5	10.64 ± 0.51 ^F^	28.89
	10	33.05 ± 1.41 ^E^	
	20	46.04 ± 2.26 ^D^	
	30	53.80 ± 2.60 ^C^	
	40	59.83 ± 3.01 ^B^	
	50	74.28 ± 3.51 ^A^	
	LSD_0.05_	1.81 ***	
Ascorbic acid	1	2.52 ± 0.01 ^F^	12.48
	2.5	10.52 ± 0.02 ^E^	
	5	36.77 ± 0.17 ^D^	
	10	49.62 ± 0.31 ^C^	
	15	59.33 ± 1.12 ^B^	
	20	69.11 ± 1.43 ^A^	
	LSD_0.05_	1.61 ***	

Values are the mean (*n* = 3) ± standard deviation. LSD_0.05_ is the least significant difference between two means, as each test was run in duplicate (calculated by factorial ANOVA). Different superscript letters within each treatment (column) express significant variation at a probability level of 0.05 (Duncan's test). *****: significant at *p* ≤ 0.001.

**Table 3 molecules-27-03177-t003:** Antibacterial activity of the methanol extract from the aerial parts of *R. vesicarius* and some selected reference antibiotics at a concentration of 10 mg/mL.

Microbes	*R. vesicarius* (10 mg/mL)	Standard Antibiotic (10 mg/L)			
		Cephradin	Tetracycline	Azithromycin	Ampicillin
Gram-negative bacteria					
*Escherichia coli*	21.07 ± 0.80 ^B^	16.37 ± 0.62 ^D^	19.61 ± 0.74 ^BC^	19.01 ± 0.72 ^B^	19.77 ± 0.75 ^C^
*Pseudomonas aeruginosa*	10.12 ± 0.38 ^E^	0.00 ^F^	0.00 ^E^	13.06 ± 0.49 ^C^	0.00 ^F^
*Salmonella Typhimurium*	23.46 ± 1.69 ^A^	0.00 ^F^	11.05 ± 0.42 ^D^	0.00 ^D^	0.00 ^F^
*Klebsiella pneumoniae*	14.38 ± 0.54 ^D^	11.55 ± 0.44 ^E^	19.09 ± 0.72 ^C^	12.44 ± 0.47 ^C^	6.08 ± 0.23 ^E^
Gram-positive bacteria					
*Bacillus cereus*	22.91 ± 0.96 ^AB^	19.67 ± 0.74 ^BC^	11.04 ± 0.62 ^D^	18.99 ± 0.82 ^B^	8.11 ± 0.31 ^D^
*Staphylococcus aureus*	17.83 ± 0.67 ^C^	20.14 ± 0.76 ^B^	21.77 ± 0.82 ^AB^	19.35 ± 0.73 ^B^	29.61 ± 1.72 ^A^
*Staphylococcus haemolyticus*	13.66 ± 0.52 ^D^	24.80 ± 1.94 ^A^	22.68 ± 0.96 ^A^	22.08 ± 0.93 ^A^	21.04 ± 0.99 ^C^
*Staphylococcus xylosus*	10.54 ± 0.40 ^E^	18.53 ± 1.70 ^C^	20.45 ± 0.77 ^ABC^	21.41 ± 0.81 ^A^	24.11 ± 1.81 ^B^
LSD_0.05_	0.0000 ***	0.0000 ***	0.0000 ***	0.0000 ***	0.0000 ***

Values are the diameters of the inhibition zone (mm) as an average of three replications ± standard error. Different superscript letters within each treatment (column) express significant variation at a probability level of 0.05 (Duncan’s test). LSD: least significant difference *** *p* < 0.001.

**Table 4 molecules-27-03177-t004:** Cytotoxic results of the extracted *R. vesicarius* on HepG2 cancer cell line.

Sample	Conc. (µg/mL)	R_1_ ^[a]^	R_2_ ^[a]^	IC_50_ (µg/mL) ^[b]^
*R. vesicarius*	1000	0.22	0.218	501.4
	500	0.92	0.96	
	125	1.6	1.6	
	62.5	1.608	1.683	
	31.3	1.63	1.64	
	0	1.3	1.3	

^[a]^ R_1_ and R_2_ specify the doublet of sample absorbance at a definite concentration; ^[b]^ IC_50_ values specify the sample’s half-maximal inhibitory concentration toward cancer cell growth.

**Table 5 molecules-27-03177-t005:** Larvicidal effect of the methanol extract from the aerial parts of *R. vesicarius* on third-instar larvaef *Aedes aegypti*.

Conc. (mg/L)	*R. vesicarius*	
	24 h Post Treatment	48 h Post Treatment
1200	28.9 ± 1.10 ^A^	42.6 ± 2.00 ^A^
1000	25.7 ± 1.00 ^B^	28.9 ± 1.10 ^B^
500	15.5 ± 1.10 ^C^	22.4 ± 1.20 ^C^
250	7.7 ± 1.10 ^D^	14.4 ± 2.20 ^D^
125	0.00 ^F^	1.1 ± 1.10 ^E^
Control	1.1 ± 1.10 ^E^	1.1 ± 1.10 ^E^
*F-*value	153.883 *	111.955 *
*p-*value	(<0.001 *)	(<0.001 *)
LC_50_	19.99	14.97
LC_90_	36.12	27.43

Mortality is expressed as the mean ± SE (standard error) of three replicates. Different superscript letters within each treatment (column) express significant variation at a probability level of 0.05 (Duncan’s test). * *p* < 0.05.

## Data Availability

The data presented in this study are available on request from the corresponding author.

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
