# Peer review of "Investigation and Biological Assessment of Rumex vesicarius L. Extract: Characterization of the Chemical Components and Antioxidant, Antimicrobial, Cytotoxic, and Anti-Dengue Vector Activity"

_molecules, 2022, doi:10.3390/molecules27103177_

Round 1
Reviewer 1 Report
The manuscript entitled "Investigation and biological assessment of Rumex vesicarius L. extract: Characterization of the chemical components, antioxidant, antimicrobial, cytotoxic and anti-dengue vector activity" presents results dealing with the phytochemical composition of a methanolic extract of the indicated plant. Some biological activities of the same extract have also been investigated. The manuscript is interesting, however it suffers from the following limitations:
- In the abstract section, what the authors mean by "... forty-four components were investigated signifying 99.99% of the total mass" ? 99.99 % of what ? of the total phytochemicals present in the methanolic extract ?
- The originality of the manuscript should be clearly indicated taking into account the investigations recently published on the same plant.
- The botanical identification of the used plant should be specified in the M & M section. A voucher specimen with a code number should added in the manuscript.
- One of the main limitations of the manuscript concern the phytochemical composition of the extract which has been achieved through GC-MS analysis technique. The obtained results of the separation as showed by the given chromatogram is very poor which let the conclusions on the identified compounds and the further discussion on the observed activities really very speculative.
- The used GC-MS which is suitable for the separation of volatile compounds and not for other ones such as polyphenols, flavonoids or others. These compounds are obviously involved in the explored activities but missed in the discussion due to the fact that they are not detected through the used analysis technique.
Reviewer 2 Report
The manuscript reports an interesting study aimed to determine the chemical compounds responsible for the biological effects of Rumex vesicarius methanol extract. In my opinion, this paper can be considered for publication in Molecules. The data collected look comprehensive and interesting. However, manuscript should be improved. Below I put my comments and suggestions.
Please correct for typing errors. For example "gas chromatography-mass spectroscopic analysis" as "gas chromatography-mass spectrometry analysis".
Page 12, line 346 - The authors do not state how they quantified the compounds identified by GC-MS analysis.
Please correct the numbering of the figures. Figure 2 appears twice.
The resolution of Figures 2 and 3 is too low.
In my opinion, the authors should also determine the profile of phenolic compounds present in the extract by HPLC-DAD or HPLC-DAD-ESI-MS/MS analysis. It is a major lack in this study because the authors in their earlier publication [15] showed that R. vesicarius shoot is rich in phenolic compounds including tannins and flavonoids.
Round 2
Reviewer 1 Report
The reviewer appreciated the efforts made by the authors to improve the quality of the manuscript, but still think that the paper could not be accepted in the present form. The main limitations are as follow:
- In their answers, he authors indicated that "we identified 99.99% of the compounds in the total mass of the extracted residues" which is not correct as only the volatile compounds are detected through GC techniques and as indicated later by the authors.
- The given chromatogram is of a poor quality for publication and showed a poor separation
- The phytochemical composition of the non volatile fraction of the extract is needed for a real scientific discussion.
Reviewer 2 Report
The authors have satisfactorily addressed most of my concerns. I have no additional comments.
